# Dual-branch differential channel hypergraph convolutional network for human skeleton based action recognition

Dong Chen [1,2], Kaichen She [1,2], Peisong Wu[1,2], Mingdong Chen[1,2], Chuanqi Li [2]*

**1** College of Physics and Electronic Engineering, Nanning Normal University, Nanning, China, **2** Key Laboratory of Functional Information Materials and Intelligent Information Processing, Nanning, China

☯ These authors also contributed equally to this work.
* lcq@mailbox.gxnu.edu.cn

## Abstract

Graph Convolutional Networks (GCNs) perform well in skeleton action recognition tasks, but their pairwise node connections make it difficult to effectively model high-order dependencies between non-adjacent joints. To address this issue, hypergraph methods have emerged with the aim of capturing complex associations between multiple joints. However, existing methods either rely on static hypergraph structures or fail to fully exploit feature interactions between channels, limiting their ability to adapt to complex action patterns. Therefore, we propose the Dual-Branch Differential Channel Hypergraph Convolutional Network (DBC-HCN), which leverages hypergraphs' ability to represent a priori non-natural dependencies in skeletal structures. It extracts spatio-temporal topological information and higher-order correlations by integrating static and dynamic hypergraphs, leveraging channel optimization and inter-hypergraph feature interactions. Our network comprises two parallel streams: a Spatio-Temporal Dynamic Hypergraph Convolutional Network (ST-HCN) and a Channel-Differential Hypergraph Convolutional Network (CD-HCN). The Spatio-Temporal Dynamic Hypergraph Convolutional stream is mainly based on the natural topology of the human skeleton, and uses dynamic hypergraphs to model the dependencies of skeletal points in spatio-temporal dimensions, so as to accurately capture the spatio-temporal characteristics of the movements. In contrast, Channel-Differential Hypergraph Convolutional stream focuses on the feature differences between different channels and extracts the characteristics of motion changes between individual skeletal points during action execution to enhance the portrayal of action details. In order to enhance the network's representational capability, we fuse the dual streams with different action feature representations, so that the Spatio-Temporal Dynamic Hypergraph Convolutional stream and the Channel-Differential Hypergraph Convolutional stream learn from each other's representations to better enrich the action feature representations. We experiment the model on three datasets, Kinetics-Skeleton 400, NTU RGB + D 60 and NTU RGB + D

**Data availability statement:** All relevant data are within the manuscript and its Supporting Information files.

**Funding:** This work was supported by Education Department of Guangxi Science and Technology Program (NO.GUIKEAB23075177) and Guangxi Zhuang Autonomous Region (Hechi University) (No. 2024GXZDSY015). These funds were both received by Drs. Dong Chen and Kaichen She.

**Competing interests:** The authors have declared that no competing interests exist.

120, and the results show that our proposed network is more competitive. The accuracy reaches 96.9% and 92.7% for the cross X-View and X-Sub benchmarks of the NTU RGB + D 60 dataset, respectively. Our code is publicly available at: https://github.com/hhh1234hhh/DBC-HCN.

---

## 1. Introduction

In the field of computer vision, action recognition[1], a core research area, has been extensively studied across various feature characterization modalities, including RGB image frames, human skeleton data, and depth maps. In recent years, human skeleton-based action recognition has drawn significant attention due to its superiority in model training for dealing with background noise and its robustness against challenges such as illumination changes, color distortions, and occlusions. In contrast to traditional action recognition methods that utilize RGB image frames, human skeletal data not only preserves high-order motion information but also integrates temporal and spatial information to effectively extract spatio-temporal action features. This capacity is essential for advancing human-computer interaction, action recognition research, and video content analysis.

Early in the development of deep learning, the prevalent methods, Recurrent Neural Networks (RNNs) and Convolutional Neural Networks (CNNs), treated human joint coordinates as independent features. These coordinates were then organized into sequences of vectors [2,3,4] or pseudo-images [5,6,7] and input into the network for action label prediction. Since this vectorization or pseudo-image representation disregarded the correlations between joints, these methods could not fully extract the features present in the skeletal data. Therefore, Yan et al. [8]used GCNs to model the correlation between human joints and graphs and proposed a spatio-temporal graph convolutional network (ST-GCN) to describe joints with a joint segmentation strategy. This static segmentation strategy, however, proved to be insufficiently adaptive to the diversity of human actions. Therefore, in 2019, Shi et al. [9] introduced a novel two-stream adaptive graph convolutional network (2s-AGCN) for skeleton-based action recognition. In contrast to traditional approaches that rely on fixed graph structures, 2s-AGCN adaptively learns the graph structure based on the input data. This means that the network can automatically adjust the connection relationships between skeletal points according to different types of actions, thus better capturing the specific features of actions. However, 2s-AGCN applies a single topology to all channels, thereby forcing the GCN to aggregate only features associated with that topology, limiting the flexibility of feature extraction. To address this limitation, Chen et al. [10] proposed the Channel Topology Refinement Graph Convolutional Network (CTR-GCN) in 2021. CTR-GCN dynamically infers channel-specific correlations, accurately capturing the relationships between vertices in each channel. This generation of channel topologies, achieved through the refinement process, eliminates the need for independent modeling of each channel's topology, significantly reducing the complexity of channel topology modeling. These advancements in GCN models optimize the

topology map structure and significantly enhance the flexibility of feature extraction. Nevertheless, GCN-based methods for skeleton-based action recognition still present two significant limitations: (1) Edges in the graph structure connect only adjacent nodes. This does not reflect dependencies between multiple joints or indirectly related joints, thus omitting significant hidden higher-order information between joints. (2) Sample skeleton graph structures are fixed, despite differences in the positions and angles of human joints across different samples in the dataset. This results in incomplete joint correlation data.

To address the aforementioned challenges, this paper proposes a Dual-Branch Differential Channel Hypergraph Convolutional Network (DBC-HCN). We utilize static and dynamic hypergraphs to model the skeleton data in DBC-HCN. The topology of the dynamic hypergraphs is constructed through K-NN and K-means algorithms [11,12], as illustrated in Fig 1. Specifically, our designed network consists of two parallel streams: a Spatio-Temporal Dynamic Hypergraph Convolutional Network (ST-HCN) and a Channel-Differential Hypergraph Convolutional Network (CD-HCN). Concerning the ST-HCN stream, in the spatial topology module, we effectively extract information regarding human joints by leveraging feature interactions among multiple hypergraphs, and further refine these hypergraphs utilizing the channel refinement module, followed by the fusion of information from these two types of information derived from the topological structure. This model-building strategy increases the flexibility of graph construction models and strengthens the correlation between joints lacking natural dependencies. In the temporal convolution module, the spatial information is processed utilizing a Dilated Temporal Convolutional Network (DTCN) to further derive the spatio-temporal features of the skeletal data. Regarding the CD-HCN stream, the primary goal is to capitalize on the characteristic feature differences across various channels to capture the inter-channel relationships between joints and employ spatio-temporal learning for effective feature fusion. Besides, considering the differing representational features of the two streams, we conduct feature fusion of the two streams to enable complementary information exchange, enriching the action features through mutual learning between the different information streams and thereby improving the accuracy of skeletal action recognition.

Our primary contributions are summarized as follows:

- We propose a novel action recognition algorithm based on Spatio-Temporal Dynamic Hypergraph Convolution, which adopts a hypergraph representation strategy with spatial channel refinement, and constructs a novel model of spatio-temporal dependencies by integrating interaction features between spatial hypergraphs and merging them with the emerging Dilated Temporal Convolution operation.

- We design the hypergraph convolutional network model with a Channel Differential Mechanism, which focuses on the frame-to-frame variability in the channel, and can effectively extract more different feature information.

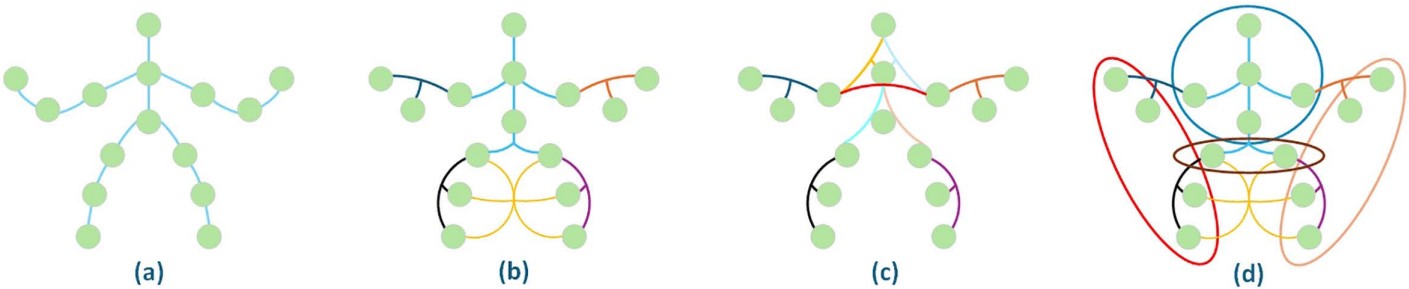

**Fig 1. Graph-based and hypergraph-based representations of the human skeleton. (a)** Representation of a traditional skeleton graph with nodes on the lines indicating human joints. **(b)** Represents a static hypergraph with six hyperedges, and different colored lines indicate different hyperedges. **(c)** Represents the hypergraph topology obtained by the K-NN method. **(d)** Denotes the hypergraph topology with global information obtained by the K-means method.

- We propose the DBC-HCN parallel network. This architecture significantly improves the effectiveness of feature extraction concerning node interactions across different channels, while preserving the ability to extract features based on the topological connections in the human skeleton graph. Extensive experimental results demonstrate that the DBC-HCN model achieves highly competitive, state-of-the-art performance on the NTU RGB+D 60, NTU RGB+D 120 datasets and Kinetics-Skeleton dataset.

## 2. Related work

In this section, we primarily review skeleton-based action recognition methods, including recurrent neural network (RNN), convolutional neural network (CNN), and graph convolutional network (GCN) methods. In addition, we provide a brief introduction to hypergraph neural networks.

### 2.1. RNN and CNN based action recognition methods

With the development of deep learning, RNN- and CNN-based action recognition methods have become the dominant way to solve the problem of skeleton-based human action recognition. RNN-based methods are able to capture long-term dependencies in the data and usually use coordinate vectors to model the skeleton data. Yong et al. [2] developed an end-to-end hierarchical RNN architecture, which has the advantage of processing time series data, enabling it to extract richer spatio-temporal features. Song et al [13] proposed an end-to-end spatio-temporal attention RNN that selectively focuses on the key spatial and temporal features to improve the understanding and recognition of complex actions. Zhang et al. [14] proposed a viewpoint-adaptive RNN with a Long Short-Term Memory (LSTM) network architecture that dynamically self-adjusts its orientation to skeletal data to the most optimal configuration in one end-to-end mode to enhance the performance of human action recognition.

In comparison, CNN-based methods are able to excellently deal with image and spatial aspects of data, and generally pseudo-images model the skeleton data. Liu et al. [15] proposed a multi-stream CNN fusion network based on a viewpoint-invariant approach for enhanced skeleton visualization, and by exploring the complementary properties between different types of enhanced color images, as a way to improve the performance of action recognition. Ke et al. [16] proposed a new method for 3D action recognition of skeletal sequences based on cylindrical surface coordinates. This method utilizes a multitask learning network to process feature vectors from all time frames in parallel for action recognition. Cao et al. [17] designed CNNs combining residual blocks and gated connectivity, which effectively utilized the ability of CNNs to process low-dimensional skeletal data and significantly improved the recognition accuracy of the network through the incorporation and improvement of residual blocks. Mumtaz et al. [18] studied a new dynamic frame skipping technique to generate meaningful temporal representations for spatio-temporal motion sequences, aiming to better detect anomalous actions. However, these RNN- and CNN-based methods cannot prove the graph structure of skeleton data well, resulting in certain missing skeleton data information when extracting features. Therefore, there is a need to find more effective methods for representing skeleton data.

### 2.2. GCN-based action recognition methods

GCN is a novel action recognition method in the field of deep learning, which utilizes graph structures to model the spatial relationships and temporal dynamics between human skeletons or joint points to efficiently capture the features of human actions. The method based on spatio-temporal graph convolutional networks (ST-GCN) was developed by Yan et al. [8]. This is the first attempt to employ GCN for modeling skeletal data. It enables the network to address spatial structure information as well as temporal dynamic information by extending graph convolution to the spatio-temporal domain, and also resolves the problem of traditional skeleton modeling that depends on handcrafted components or traversing rules, thereby enhancing the generalization and expressive power of the network. Shi et al. [9] have put

forward a new type of two-stream adaptive graph convolutional network (2s-AGCN). It defines a dual-stream network with respect to spatial and temporal streams employing the adaptive graph convolution concept. This enhances the degree of freedom of graph models constructed and their applicability to a wide range of datasets. Ye et al. [19] proposed a novel Dynamic GCN (Dynamic GCN), an architecture that incorporates a Context Encoding Network (CeN) designed to learn context-rich dynamic skeleton topologies. CeN enables the modeling of dynamic graph topologies by integrating contextual information from other joints into the relationship between any two joints. Accordingly, the model is able to identify slight differences in the actions, resulting in improved performance in action recognition and enhancing the generalization ability of the model. Plizzari et al. [20] proposed a new Spatio-Temporal Transformer Network (ST-TR), which uses the Transformer's Self-Attention Operator to capture dependencies between articulation points. The model effectively extracts the spatio-temporal features of the skeleton data in space and time through the Transformer architecture, which thus optimizes the performance related to the skeletal action recognition task. Song et al. [21] put forth a collection of efficient GCN baseline models (Efficient GCN) that incorporate increased accuracy while having a small number of trainable parameters, create stronger models through a spatio-temporal separation learning strategy, achieve faster baselines, and, further, lower the modeling complexity and over-parameterization. Nevertheless, despite the superiority of such models in performing on simple data relationships derived from pairwise interaction as well as graph neural networks, they exhibit limitations when dealing with higher-order data relationships and more complex structures.

### 2.3. Hypergraph neural networks

Hypergraph neural network (HGNN) [22] is a state-of-the-art data representation learning method designed to capture higher-order dependencies among data points. Unlike traditional graph neural networks, HGNN connects an arbitrary number of nodes via hyper edges, thus enabling the modeling of more complex data relationships than pairwise connections. Jiang et al. [23] proposed the Dynamic Hypergraph Neural Network (DHNN), which is capable of updating the hypergraph structure at each layer of the network to adapt to the dynamics of the action. Moreover, Gao et al. [24] developed a dynamic hypergraph representation and learning framework which effectively portrays the higher-order dependencies between nodes in the hypergraph structure through tensor representations, providing an important extension to existing hypergraph learning techniques. Mei et al. [25] proposed a Dynamic Hypergraph Hyperbolic Neural Network (DHHNN) based on variational autoencoder, which integrates hyperbolic geometry, dynamic hypergraph structure, and self-attention mechanism to dynamically adjust the contribution degrees of nodes and hyperedges, effectively enhancing the modeling capability for complex multimodal data. Overall, the hypergraph neural network through its unique structure, is able to effectively extract and represent higher-order features in skeleton data, thus demonstrating excellent performance in skeleton-based action recognition tasks.

## 3. Method

In this section, we begin with a concise introduction to Graph Convolutional Networks (GCNs) and Hyper Graph Convolutional Networks (HGCNs). We then demonstrate the DBC-HCN and present detailed information on the complete structure of the network.

### 3.1. Graph convolutional networks

In traditional GCNs, a skeleton sequence consists of $T$ single-frame skeleton graphs. The data is represented by a graph structure $G = (V, E)$, where $V$ represents the set of joints and $E$ represents the set of edges. These edges include the skeletal edges $E_S$, which connect corresponding nodes in each frame, and the edges $E_T$, which link skeletal points across all frames. The feature matrix of the skeleton graph is represented by the adjacency matrix $A \in R^{N \times N}$, whose element values reflect the connectivity between joints. The update rule of the graph convolutional network is defined as follows:

$$X^{(l+1)} = \sigma\left(\widetilde{D}^{-\frac{1}{2}}\widetilde{A}\widetilde{D}^{-\frac{1}{2}}X^{(l)}W^{(l)}\right) \tag{1}$$

where $X^{(l)} \in R^{N \times C}$ is the input signal of the convolutional layer $l$, and $X^{(0)}$ denotes the initial feature of the input layer, $W^{(l)} \in R^{C \times C}$ is the learnable weight matrix of the convolutional layer $l$, $\sigma(\cdot)$ refers to the nonlinear activation function, $\widetilde{A}$ is the normalized graph adjacency matrix, and $\widetilde{D}$ is the degree matrix of $\widetilde{A}$, while $N$ is the number of nodes in the graph, and $C$ is the node vector dimension.

### 3.2. Hypergraph convolutional networks

A hypergraph is defined by a triple $G_h = (V_h, \xi_h, W_h)$ consisting of a node set $V_h$, a hyperedge set $\xi_h$, and a hyperedge weight set $W_h$, whose hyperedges are capable of connecting an arbitrary number of nodes for a more thorough representation of relationships. For the hypergraph $G_h$, the association matrix $H$ is shown in Fig 2 and its elemental relations are defined as:

$$H = \begin{cases} h(v, e) = 1 & v \in e \\ h(v, e) = 0 & v \notin e \end{cases} \tag{2}$$

The degree of a given node $v \in V_h$ denotes the number of hyperedges containing that node, denoted as:

$$d(v) = \sum_{e \in \xi_h} W_h(e)h(v, e) \tag{3}$$

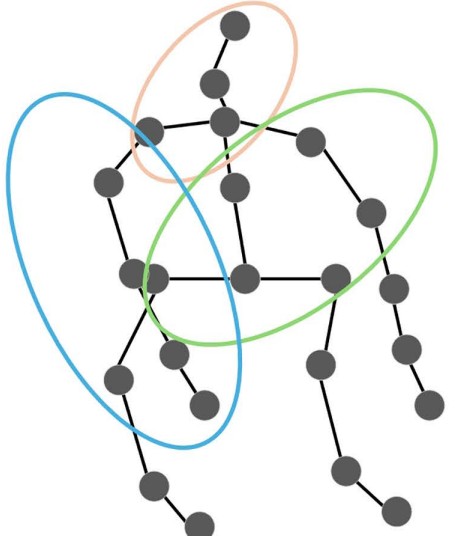

| | e1 | e2 | e3 | · · · | en |
|---|---|---|---|---|---|
| V1 | 1 | 0 | 0 | · · · | |
| V2 | 0 | 0 | 1 | · · · | |
| V3 | 0 | 1 | 0 | · · · | |
| ⋮ | ⋮ | ⋮ | | | |
| Vn-1 | 1 | 0 | 0 | · · · | |
| Vn | 0 | 0 | 1 | · · · | |

(a)                                        (b)

**Fig 2. (a) Representation of a hypergraph convolutional network.** Grey circles indicate different nodes and closed lines indicate different hyperedges. (b) Representation of the feature matrix of the hypergraph convolutional network.

where $W_h(e)$ is the weight of the hyperedge $e$ and the degree of the hyperedge $e \in \xi_h$ denotes the number of joints that constitute the hyperedge $e$, denoted as:

$$\delta(e) = \sum_{v \in V_h} h(v, e)$$

(4)

The update rule for hypergraph convolutional networks is defined as [24]:

$$X^{(l+1)} = \sigma(D_v^{-\frac{1}{2}} HWD_e^{-1} H^T D_v^{-\frac{1}{2}} X^{(l)} \Theta^{(l)})$$

(5)

where $\Theta^{(l)} \in R^{C \times C}$ is the weight matrix of the convolutional layer $l$ used to extract features for the nodes in the hypergraph. $D_v$ denotes the diagonal matrix of the node degree $d(v)$, $D_e$ indicates the diagonal matrix of the hyperedge degree $\delta(e)$, and $W$ is the diagonal matrix of the weights of the hyperedge.

### 3.3. Dual-branch differential channel hypergraph convolutional network

As presented in Fig 3(a), our proposed DBC-HCN model has two parallel processing streams, namely the ST-HCN stream and the CD-HCN stream. Spatio-Temporal Dynamic Hypergraph Convolution consists of spatial convolution module

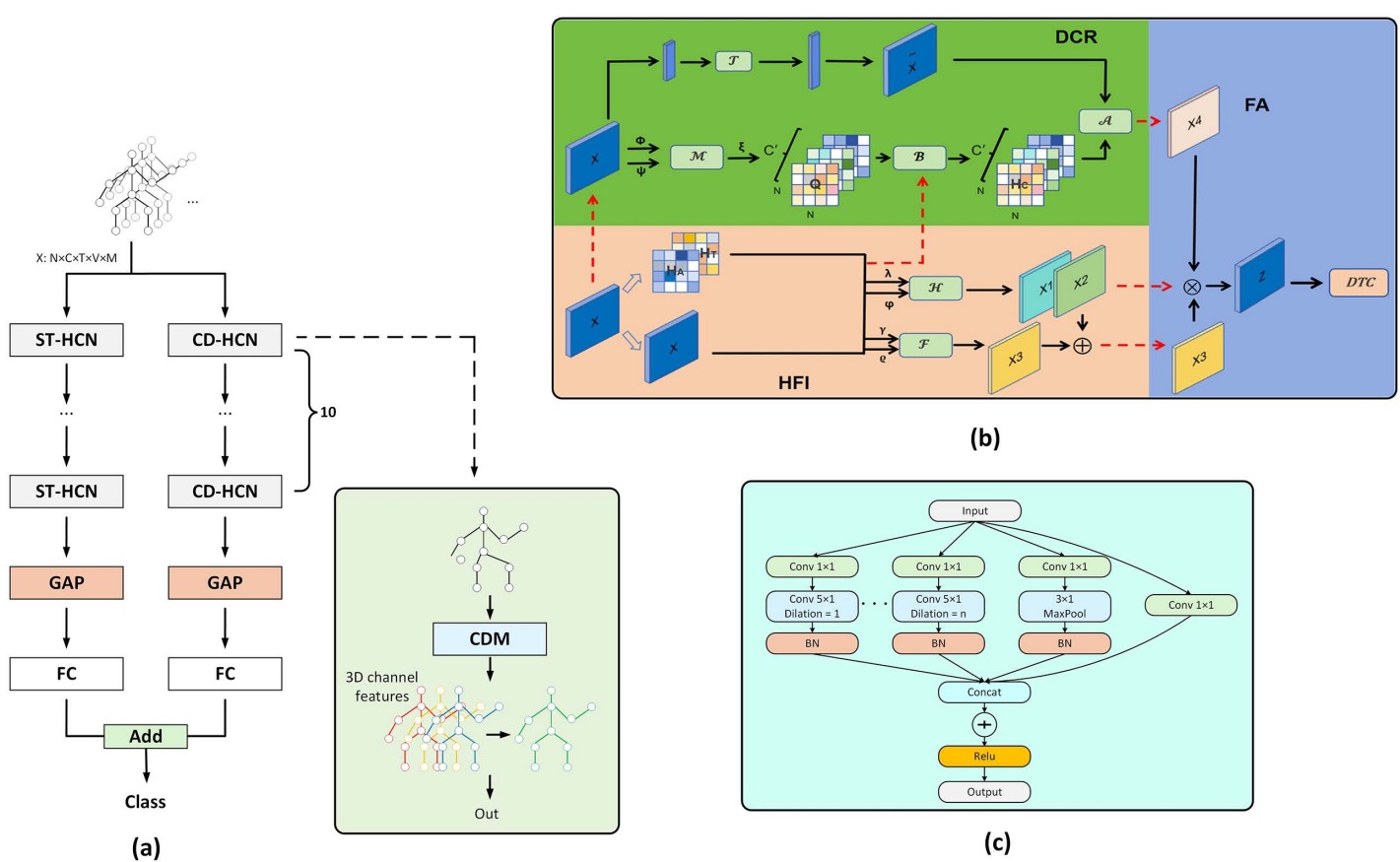

**Fig 3. (a)** The overall framework of DBC-HCN, the structure of CD-HCN and ST-HCN is similar, and the core difference is that CD-HCN uses a Channel Differential Mechanism(CDM) to extract differential features from the 3D channels of the input data. **(b)** Spatial Convolution Structure of DBC-HCN. **(c)** Dilated Temporal Convolution Structure of DBC-HCN.

composed of a Dynamic Channel Refinement (DCR) module, a Hypergraph Feature Interaction (HFI) module, and a Feature Aggregation (FA) module, and a Dilated Temporal Convolution (DTC) module. Specifically, DCR module adjusts the weights of the feature channels through an adaptive mechanism to extract key features that are closely related to action recognition. HFI module extracts the representational differences with the help of the topologies of multiple hypergraphs, and FA is the fusion of information from both sides. DTC module has the purpose of widening the sensory field of the model such that the long-term dependencies that exist in the time-series data can be captured. Compared with the ST-HCN stream, a distinctive feature of CD-HCN stream is that the inter-channel difference operation is performed on the input feature maps, and an advanced representation of the differential features is learnt by feeding the obtained differential features into the convolutional network, which ultimately improves the discriminative power of the model.

**3.3.1. Spatio-temporal dynamic hypergraph convolution.** In this section, we explore the structural details of the spatial and Dilated Temporal Convolution (DTC) in the ST-HCN, and analyze the mathematical and architectural levels of HFI module, DCR module, and FA module, which make up the spatial convolution of the model.

Hypergraph Feature Interaction (HFI). Feature interactions between hypergraphs aim to reveal the deep connections between nodes in different hypergraph structures in order to understand and utilize the information of multiple hypergraph structures. To this end, we propose a novel feature interaction mechanism as shown in Fig 3(b). First, we apply linear transformation functions $\lambda$ and $\varphi$ to the input features, followed by characterization extraction via the Einstein summation convention function $\mathcal{A}(\cdot)$, as shown in Eq:

$$X1 = \mathcal{A}\left(\lambda\left(x_i\right), H_A\right) \tag{6}$$

$$X2 = \mathcal{A}\left(\varphi\left(x_i\right), H_T\right) \tag{7}$$

where $H_T$ is the adjacency matrix of the static hypergraph, $H_N$ and $H_M$ denote the hypergraph adjacency matrices constructed by the K-NN and K-means algorithms [11,12,26], and $H_A$ represents the composite hypergraph adjacency matrix formed by the splicing operation of the adjacency matrices $H_N$ and $H_M$. $X1$ and $X2$ are the output features after the interaction of inputs and hypergraph adjacency matrices $H_A$ and $H_T$ features.

Next, an element-by-element subtraction of the feature vectors of each corresponding node in different hypergraphs is performed by the $\gamma(\cdot)$ function to obtain a difference feature matrix, Eq:

$$H_B = \gamma\left(H_A, H_T\right) = \sigma\left(H_A - H_T\right) \tag{8}$$

where $\sigma(\cdot)$ is the nonlinear activation function and $H_B$ denotes the characteristic interaction matrix of the hypergraph.

After that, we first transform the features utilizing the linear function $\varrho$, then feed the transformed features into the function $\mathcal{F}(\cdot)$ to complete the feature interaction with the interaction matrix $H_B$, and finally use the interaction features to fuse the features $X2$ as a way to obtain a more comprehensive interaction output, which is given by Eq:

$$X3 = \mathcal{F}\left(H_B, \varrho\left(x_i\right), X2\right) = \mathrm{A}\left(H_B, \varrho\left(x_i\right)\right) + \alpha \cdot X2 \tag{9}$$

where $\alpha$ is a trainable scalar parameter and $X3$ denotes the final output feature that contains the interaction information of all the above input features.

**Dynamic Channel Refinement (DCR).** We segment the Dynamic Channel Refinement process into three stages. First, the dynamic modeling and channel refinement are accomplished by inferring the hypergraph structure. Then, the transformation function is used for feature transformation. Finally, the channel topology is aggregated. In this process, $Q \in R^{N \times N \times C'}$ denotes the channel correlation matrix and $\mathcal{B} \in R^{N \times N \times C'}$ denotes the channel aggregation matrix.

 

(i) Dynamic modeling: Depicted in Fig 3(b), we begin by mapping the input features utilizing the linear transformation functions $\phi$ and $\psi$, after which these dimensionality-decreasing features are fed into the dynamic modeling function $\mathcal{M}(\cdot)$. The dynamic modeling function is denoted as:

$$\mathcal{M}\left(\phi\left(x_i\right), \psi\left(x_j\right)\right) = \sigma\left(\phi\left(x_i\right) - \psi\left(x_j\right)\right) \tag{10}$$

where $\sigma(\cdot)$ is a nonlinear activation function, and $\mathcal{M}(\cdot)$ centers on computing the distance between the node features $\phi\left(x_i\right)$ and $\psi\left(x_j\right)$ in the hypergraph along the channel dimensions and generating channel-specific representations of the topological relationships.

Next, we use a linear transformation $\xi$ on top of the modeling function to lift the features and learn the inter-channel correlation $Q \in R^{N \times N \times C'}$, Eq:

$$Q = \xi\left(\mathcal{M}\left(\phi\left(x_i\right), \psi\left(x_j\right)\right), H_T\right) \tag{11}$$

where $H_T$ is the adjacency matrix of the static hypergraph and the $\xi$ function is a multidimensional tensor algorithm.

Finally, to further refine the correlation between channels, we refine $H_T$ with channel correlation $Q$, computed as:

$$\mathcal{B} = \mathcal{A}\left(Q, H_T\right) \tag{12}$$

where $\mathcal{B} \in R^{N \times N \times C'}$ is denoted as the channel topology after the refinement process, and $\mathcal{A}(\cdot)$ is the Einstein summation convention.

(ii) Feature Transforms: we use feature transforms aiming to transform the input into a high-level feature representation via $\mathcal{T}(\cdot)$ as shown in Fig 3(b). The graph convolution with a simple linear variation is used for this purpose with the following formula:

$$\widetilde{X} = \mathsf{T}(X) = XW \tag{13}$$

where $\widetilde{X} \in R^{N \times C'}$ is the transformed feature, $X$ denotes the input feature matrix, and $W \in R^{C \times C'}$ denotes the shared weight matrix, which is responsible for linearly combining the input features during graph convolution to extract a feature representation rich in structural information.

(iii) Channel aggregation: under the premise of having channel topology $\mathcal{B}$ and high-level features $\widetilde{X}(c \in \{1, \cdots, c'\})$, the final output features $X4$ can be obtained by aggregating the channel graph through the aggregation function with the formula:

$$X4 = \mathcal{A}\left(\mathcal{B}, \widetilde{X}\right) = \sum_{c=1}^{c'} \mathcal{B}_c \cdot \widetilde{x}_{:,c} \tag{14}$$

where $\mathcal{B}_c$ and $\widetilde{x}_{:,c}$ come from the $c$-th channel of $\mathcal{B}$ with $\widetilde{X}$, the channel aggregation function $\mathcal{A}(\cdot)$ according to the given channel topology $\mathcal{B}$ and high-level features $\widetilde{X}$ obtains the output matrix $X4$ by channel aggregation of all connected channel graphs.

**Feature Aggregation (FA).** In this study, each of the four output features involved characterizes differentiated action characteristics among nodes in the network, and we aggregate the features as a way to comprehensively extract the relevance of these nodes and thus construct a more comprehensive global feature representation. The corresponding formula is:

$$X = Cat(X1, X2, X3, X4) \tag{15}$$

where *Cat* denotes the splicing operation.

Finally, on the basis of spatial convolution, we introduce a temporal convolution method for feature extraction for time series data, as shown in Fig 3(c). This study adopts the Dilated Temporal Convolution (DTC) strategy, which effectively improves the ability of feature extraction for time series data by flexibly adjusting the dilation coefficient.

**3.3.2. Channel-differential hypergraph convolution.** In order to recognise and understand human movements more accurately, we propose a Channel-Differential Hypergraph Convolution method. The method is structurally similar to ST-HC, but adopts a novel Channel Differential Mechanism (CDM). The core idea of this mechanism is to extract the dynamic changes between channels, and thus to characterise the relative motion of skeletal points in a finer way. Compared with the traditional HCN and the ST-HCN proposed above, this method exhibits significant differences in feature extraction and information modelling, enabling the model to capture the dynamic changes between channels during the action more effectively.

In most hypergraph-based action recognition methods, HCN mainly relies on absolute coordinate information of skeletal points, such as X,Y,Z 3D coordinates. However, using only absolute coordinates may not be able to adequately portray the dynamic features of human motion.ST-HCN models the higher-order relationships of skeletal points by constructing spatio-temporal hypergraphs, but it still fails to explore the channel dimension in depth. As shown in Fig 4, Channel-Differential Hypergraph Convolutional Network (CD-HCN) is able to capture the subtle changes between neighbouring time frames by introducing a Channel Differential Mechanism, which enhances the representation of the dynamic features of the movement and thus improves the accuracy of recognition.

In our approach, the core of the channel differential hypergraph convolution lies in first performing the difference operation on each channel of the input data, so as to realize the refined extraction of the features of the skeletal points, and then inputting the features into the model. Specifically, each skeletal point has a corresponding channel definition in the CD-HCN, and these channels represent the 3D coordinate information of the skeletal point respectively. By performing the difference operation on these channels, we obtain multiple single channels containing the 3D motion information of the skeletal points. This processing procedure is formally described by the following formula:

$$D_{:,i} = \frac{1}{C-1} \sum_{j=1, j\neq i}^{C} X_{:,i} - X_{:,j} \tag{16}$$

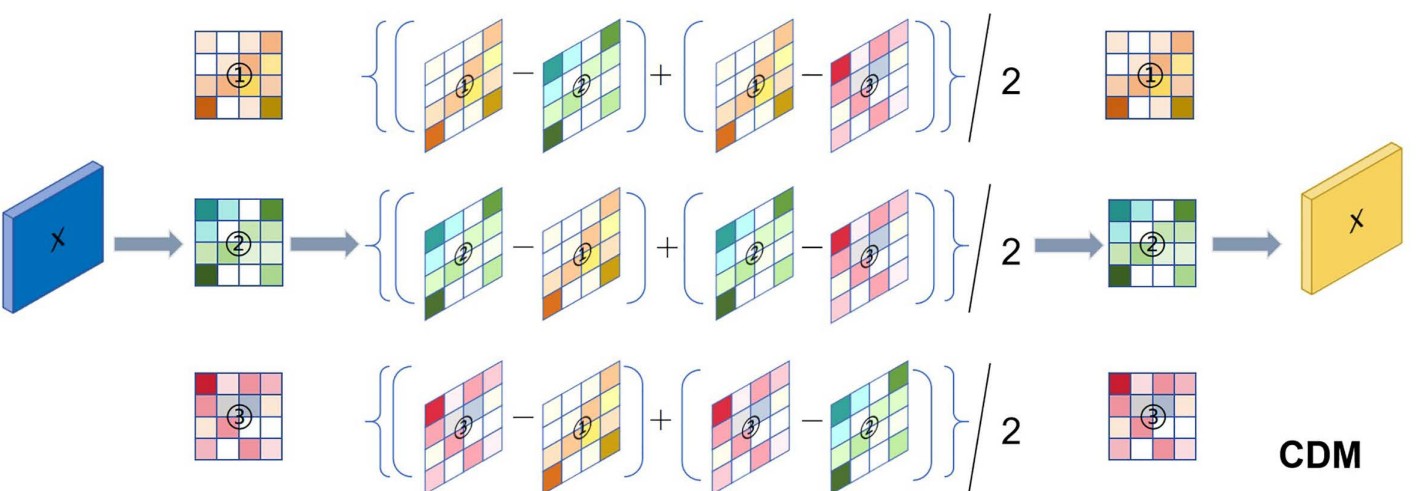

**Fig 4. Channel differential mechanism in channel-differential hypergraph convolutional network.**

where $C$ is the number of input feature channels, $X_{:,i}$ and $X_{:,j}$ are the feature vectors of the $i$-th and $j$-th channels, and $D_{:,i}$ is the difference vector of the $i$-th channel.

Compared with the traditional HCN that deals with the absolute coordinates of skeletal points directly, CD-HCN extracts neighbouring time frame change information via the Channel Differential Mechanism, enabling the model to focus on relative displacements during movement. This approach can eliminate the static differences between different individuals and improve the robustness of the model to complex movement patterns.

### 3.4. Model architecture

The architecture of the model proposed in this paper is shown in Fig 5, which consists of a two-stream network, where the second stream specifically integrates a Channel Differential Mechanism that focuses on analyzing frame-to-frame variations between channels. Both streams consist of ten layers, which include spatial convolution and temporal convolution. Spatial convolution aims to capture spatial information of joints and bones, while temporal convolution is used to extract temporal information in different frames. In spatial convolution, we incorporate Hypergraph Feature Interaction and Dynamic Channel Refinement to analyze static and dynamic hypergraph features. The convolution kernel size for temporal convolution is fixed to 5 × 1, except for the maximum pooling layer which is 3 × 1, and employs varying dilation rates to expand the receptive field. At layers 5 and 8, the temporal dimension is halved through stride temporal convolution. Subsequently, the model predicts action labels by global average pooling and FC layers.

Finally, in order to fuse the information of the two streams for interaction, we use a late fusion strategy. Specifically, the features of ST-HCN stream are integrated with CD-HCN stream, Eq:

$$O = \alpha \cdot L(V) + \beta \cdot N(V) \tag{17}$$

where $\alpha$, $\beta \in [0, 1]$, $V$ denotes human action data, and $\mathcal{L}$ and $\mathcal{N}$ denote spatio-temporal hypergraphic convolutional network and channel differential convolution, respectively.

## 4. Experiments

In this section, in order to evaluate the performance of the proposed model, we perform a number of experiments on the datasets NTU-RGB + D 60, NTU-RGB + D 120 and Kinetics-Skeleton. First, we provide a detailed description of the three

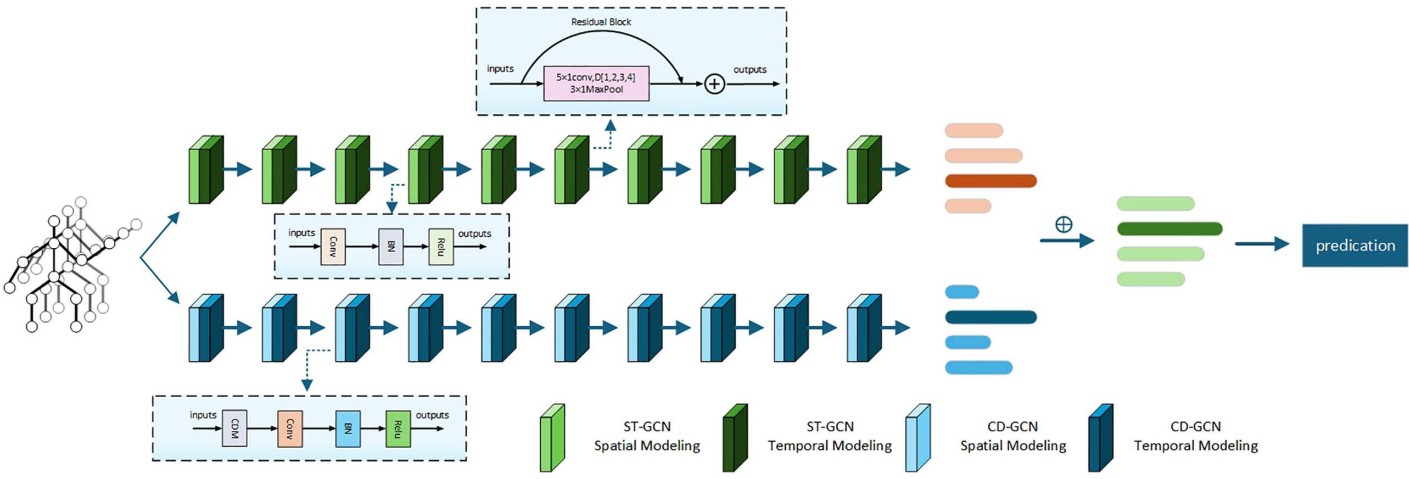

**Fig 5. The overall architecture of DST-HCN.** The temporal modeling structures of ST-GCN and CD-GCN are similar.

datasets. Then we perform an ablation study on the NTU-RGB + D 60 dataset to examine the contribution of each module to the network. Finally, we validate the effectiveness of the proposed model by comparing it with several state-of-the-art methods.

## 4.1. Datasets

**NTU RGB + D 60:** NTU RGB + D 60 [27] dataset is a large-scale dataset widely used in the field of action recognition, which contains 56,880 skeleton action sequences performed by 40 volunteers covering 60 different action classes. Each action sample ensures that a maximum of two subjects are involved and is captured simultaneously by cameras from three different viewpoints, thus providing rich information about the 3D skeletal joint points. The authors of this dataset recommend two evaluation benchmarks: (1) Cross-subject (X-Sub) benchmark: the dataset is divided into two groups, with the training data coming from 20 subjects and the test data coming from the remaining 20 subjects, for a total of 40,320 training samples and 26,560 test samples. (2) Cross-View (X-View) benchmark: training samples come from camera views 2 and 3, totaling 37,920, while test samples come from camera view 1, totaling 18,960.

NTU RGB + D 120: NTU RGB + D 120 [28] dataset is a significant extension of the NTU RGB + D 60 dataset with the addition of 57,367 new skeleton sequences and 60 new action categories, making it the largest 3D co-annotated human action recognition dataset available. The dataset consists of more than 114,000 skeleton action sequences executed by 40 volunteers in 32 different settings, each representing a different location and context, to enhance the model's ability to generalize across different environments. In order to evaluate the performance of the model, NTU-120 proposes two benchmark evaluation methods:(1) Interdisciplinary (X-Sub) Benchmark: same as the X-Sub benchmark of NTU-60, the X-Sub evaluation divides the dataset into two groups, one for training and the other for testing. (2) Cross-Set (X-Set): X-Set evaluation over splits the training and testing samples based on the name of the camera setup ID.

Kinetics-Skeleton: Kinetics-Skeleton [29] dataset is a large-scale human behavior recognition benchmark based on YouTube videos, containing about 300,000 video clips covering up to 400 human behaviors involving daily activities, motion scenarios, and complex human-computer interactions. The original Kinetics dataset only provides raw video clips without skeleton sequences.ST-GCN [2] obtains the positions of 18 joints on each frame of the video by applying the publicly available Open-Pose toolkit, and accordingly selects the two individuals with the highest average joint confidence for skeleton data extraction. The processed skeleton data were divided into 240,000 training clips and 20,000 validation clips.

## 4.2. Implementation details

All experiments were conducted utilizing the PyTorch deep learning framework, in which we implemented stochastic gradient descent (SGD) with momentum of 0.9 as the optimizer, and selected cross-entropy as the loss function for the back-propagated gradient, with weight decay configured at 0.0001. During the training phase on NTU RGB + D 60 [27] and NTU RGB + D 120 [29], the learning rate was increased at the 35th and 55th epoch, with the training process reaching completion at the 80th epoch. For Kinetics-Skeleton [22], the learning rate demonstrated decreases at the 45th and 55th epochs, and the training process reached completion at the 70th epoch. For the NTU RGB + D 60 and NTU RGB + D 120 datasets, the batch size is set to 70. For the Kinetics-Skeleton dataset, we set the batch size to 64.

## 4.3. Ablation study

In this section, to validate the effectiveness of our proposed module and dual-stream framework, we conduct the following experiments using the X-Sub benchmark on the NTU RGB + D 60 dataset.

In multi stream fusion, we integrate different output features, specifically J, B, and M (including joints, bones, and motions). In order to evaluate the effect of each stream on the model performance, we implemented three sets of controlled experiments, as shown in Table 1. The results showed that the B (bones) stream had the most significant impact, with a performance gain of up to 2.6%, thereby further improving the recognition accuracy.

**Table 1. Comparison of performance with different input data.**

| Methods | Accuracy |
| --- | --- |
| DBC-HCN (Joint) | 90.6 |
| DBC-HCN (Bone) | 91.6 |
| DBC-HCN (Motions) | 89.9 |
| DBC-HCN | 92.7 |

In order to validate the effectiveness of the DCR module, HFI module, and the dual-stream methods of ST-HCN and CD-HCN, we implemented a removal-by-removal ablation study to analyze the DBC-HCN model. By monitoring the performance changes in the B-stream, we evaluate the effectiveness of these modules, as shown in Table 2. In the DBC-HCN w/o HFI model with HFI removed, we observe a 0.6% performance degradation, thus confirming the critical role of HFI between channels. In addition, the DBC-HCN w/o DCR model shows a 2.0% performance degradation, which highlights the importance of DCR in improving model performance. Meanwhile, the performance of DBC-HCN w/o CD with removal of CD-HCN and DBC-HCN w/o ST model with removal of ST-HCN decreased by 1.4% and 1.2%, respectively, indicating the vital role of the dual-stream interaction in improving model performance.

## 4.4. Comparative study

This section presents a comparative analysis of the DBC-HCN model's performance against state-of-the-art skeleton-based action recognition methods. We will test it on different benchmarks on the NTU RGB + D 60, NTU RGB + D 120, and Kinetics-Skeleton datasets.

Comparative data for the models are shown in Tables 3, 4 and 5. Across all three datasets, our method proves superior to the vast majority of methods when evaluated against almost all metrics. Specifically, regarding the NTU RGB + D 120 dataset, our model, which combines joint and bone information, reaches state-of-the-art performance levels. The DBC-HCN model surpasses the current hypergraph model DST-HCN [37] on two evaluation benchmarks, indicating improvements of 0.6% and 0.5%, respectively.

We used the X-Sub benchmark on the NTU RGB + D 60 dataset as a challenging testbed for evaluating model performance. Fig 6 illustrates the confusion matrix of the DBC-HCN model on the NTU RGB + D 60 X-Sub benchmark. Fig 6 shows the confusion matrix of the DBC-HCN model at NTU RGB + D 60. By analyzing the confusion matrix based on X-Sub, we can see that although some classes are significantly less accurate than others, our method can accurately identify most of them. Fig 7 further evaluates this trend. BCD-HCN has an accuracy rate of over 90% in 38 action categories, accounting for 63.33% of all action categories, indicating that the model can accurately identify actions in most cases. In addition, the model has an accuracy rate of over 80% for 52 action categories, accounting for 86.66% of all action categories. These statistical data highlight the outstanding accuracy and reliability of BCD-HCN in action recognition tasks.

**Table 2. Comparison of performance when adding or removing HFI, DCR, ST-HCN or CD-HCN from DBC-HCN.**

| Methods | Params | Accuracy |
| --- | --- | --- |
| DBC-HCN w/o HFI | 2.86M | 91.0 |
| DBC-HCN w/o DCR | 3.04M | 89.6 |
| DBC-HCN w/o ST | 1.59M | 90.4 |
| DBC-HCN w/o CD | 1.58M | 90.2 |
| DBC-HCN | 3.21M | 91.6 |

**Table 3. Performance comparison of the state-of-the-art methods of DBC-HCN on the NTU RGB+D 60 dataset.**

| Methods | NTU RGB+D 60 | |
| --- | --- | --- |
| | X-Sub | X-View |
| ST-GCN [8] | 81.5 | 88.3 |
| 2s-AGCN [9] | 88.5 | 95.1 |
| Shift-GCN [30] | 90.7 | 96.5 |
| Dynamic GCN [19] | 91.5 | 96.0 |
| MS-G3D [31] | 91.5 | 96.2 |
| ST-TR [20] | 89.3 | 96.1 |
| MTSA-GCN [32] | 90.8 | 96.7 |
| Hyper-GNN [33] | 89.5 | 95.7 |
| DHGCN [34] | 90.7 | 96.6 |
| SpSt-GCN [35] | 91.6 | 95.8 |
| SA-TDGFormer [36] | 92.7 | 96.8 |
| DBC-HCN(Ours) | 92.7 | 96.9 |

**Table 4. Performance comparison of the state-of-the-art methods of DBC-HCN on the NTU RGB+D 120 dataset.**

| Methods | NTU RGB+D 120 | |
| --- | --- | --- |
| | X-Sub | X-Set |
| ST-GCN [8] | 70.7 | 73.2 |
| 2s-AGCN [9] | 82.5 | 84.2 |
| Shift-GCN [30] | 85.9 | 87.6 |
| Dynamic GCN [19] | 87.3 | 88.6 |
| CTR-GCN [10] | 88.9 | 90.6 |
| DHGCN [34] | 86.0 | 87.9 |
| EfficientGCN-B4 [21] | 88.7 | 88.9 |
| DST-HCN [37] | 88.8 | 90.7 |
| SpSt-GCN [35] | 87.8 | 88.8 |
| SA-TDGFormer [36] | 86.8 | 88.9 |
| DBC-HCN(Ours) | 89.4 | 91.2 |

**Table 5. Performance comparison of state-of-the-art methods for DBC-HCN on the Kinetics-Skeleton dataset.**

| Methods | Kinetics-Skeleton | |
| --- | --- | --- |
| | Top1 | Top5 |
| ST-GCN [8] | 30.7 | 52.8 |
| 2s-AGCN [9] | 36.1 | 58.7 |
| ST-TR [20] | 37.4 | 59.8 |
| Hyper-GNN [33] | 37.1 | 60.0 |
| DHGCN [34] | 37.7 | 60.6 |
| DBC-HCN(Ours) | 38.4 | 61.4 |

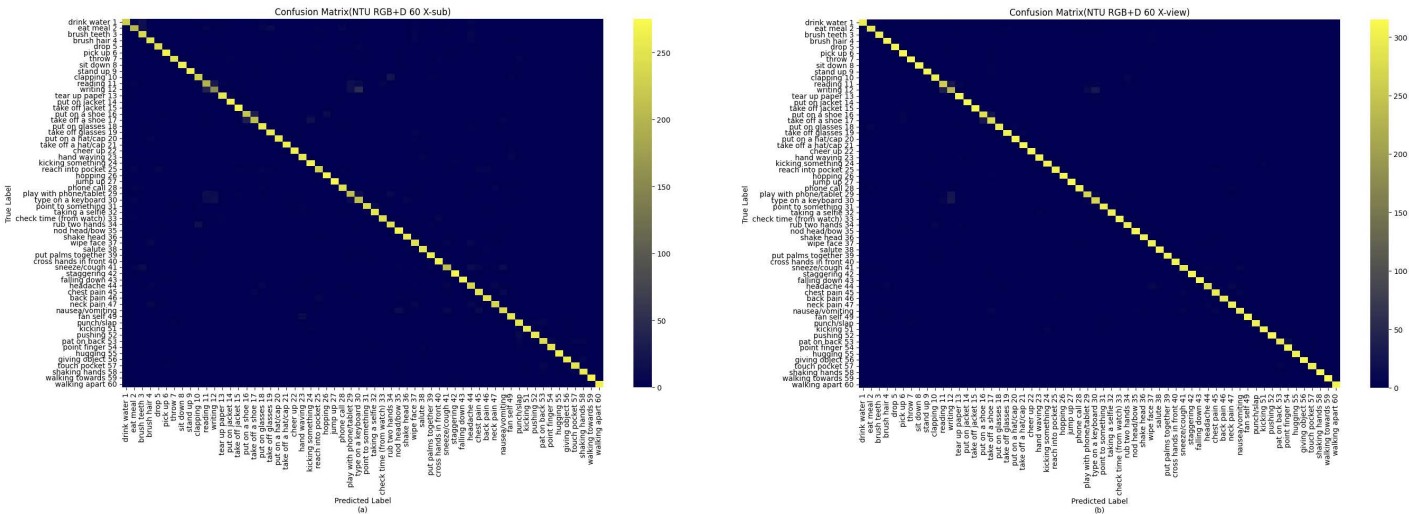

**Fig 6. Confusion matrix for the NTU RGB+D 60 dataset.** The yellower the square on the diagonal, the more accurate the identification. **(a)** X-sub benchmark on the NTU RGB+D 60 dataset. **(b)** X-view benchmark on the NTU RGB+D 60 dataset.

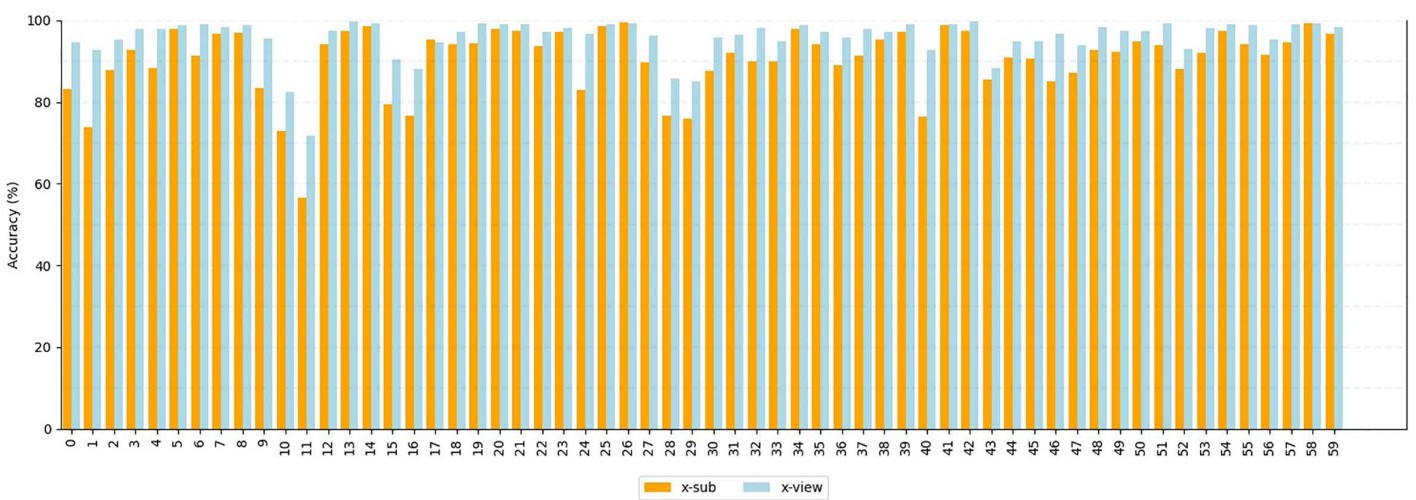

**Fig 7. The accuracy of x-sub and x-view in NTU RGB+D 60 for this work.**

Among the 60 action categories, the model demonstrates particularly strong performance in recognizing actions such as "take off jacket", "jump up", "hopping", "staggering" and "walking towards". We can attribute the high accuracy for these actions to their unique visual features and the model's successful capture of these characteristics. However, the Staple action recorded the lowest recognition accuracy.

## 5. Conclusion

This paper proposes the Dual Branch Differential Channel Hypergraph Convolutional Network (DBC-HCN), which improves the performance of skeleton-based action recognition by integrating Spatio-Temporal Dynamic Hypergraph Convolution and Channel-Differential Hypergraph Convolution. Compared with traditional graph convolution methods, this

model utilizes a hypergraph structure to process high-order correlations of skeleton points, effectively capturing complex spatiotemporal features in motion, and enhancing action detail representation through Dynamic Channel Refinement module, Hypergraph Feature Interaction module, and Channel Difference Mechanism. The experiment shows that DBC-HCN performs better than mainstream methods on multiple datasets, verifying its superiority in modeling complex actions.

Although the performance of the model is excellent, it still has dual limitations: On the one hand, the dual branch architecture and hypergraph computing mechanism lead to a significant increase in computational complexity, which restricts the real-time deployment capability on edge computing devices; On the other hand, the construction of dynamic hypergraphs relies on K-NN and K-means algorithms, which limit their generalization ability to unconventional action patterns. In response to the above issues, future research will advance in three directions: firstly, developing a lightweight hypergraph convolution framework, combining network pruning and quantization techniques to achieve efficient inference at the edge; Secondly, an attention driven mechanism is introduced to optimize the weight allocation of hyperedges, enhancing the robustness of modeling heterogeneous action patterns; Thirdly, the engineering validation of the final extended model in practical scenarios such as medical rehabilitation abnormal action detection and intelligent human-computer interaction.

## Supporting information

**S1 File. Code for key modules.** It briefly contains the code of the key modules of the neural network in this experiment. (PDF)

## Author contributions

**Supervision:** Chuanqi Li.

**Validation:** Peisong Wu, Mingdong Chen, Chuanqi Li.

**Writing – original draft:** Dong Chen, Kaichen She.

**Writing – review & editing:** Dong Chen, Kaichen She.

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
