## [Decision Letter · Decision Letter 0]

11 Jun 2025

Dear Dr. Li,

Thank you for submitting your manuscript to PLOS ONE. After careful consideration, we feel that it has merit but does not fully meet PLOS ONE’s publication criteria as it currently stands. Therefore, we invite you to submit a revised version of the manuscript that addresses the points raised during the review process.

We look forward to receiving your revised manuscript.

Kind regards,

Xiyu Liu

Academic Editor

PLOS ONE

Journal Requirements:

Key Laboratory of AI and Information

Processing, Education Department of Guangxi Zhuang Autonomous Region (Hechi University) (No. 2024GXZDSY015),Guangxi Science and Technology Program  (2023AB29003 and  2023AB01005).

This work was supported by Key Laboratory of AI and Information Processing, Education Department of Guangxi Zhuang Autonomous Region (Hechi University) (No. 2024GXZDSY015), Guangxi Science and Technology Program (2023AB29003 and 2023AB01005).

Key Laboratory of AI and Information

Processing, Education Department of Guangxi Zhuang Autonomous Region (Hechi University) (No. 2024GXZDSY015),Guangxi Science and Technology Program  (2023AB29003 and  2023AB01005).

6. Please amend your list of authors on the manuscript to ensure that each author is linked to an affiliation. Authors’ affiliations should reflect the institution where the work was done (if authors moved subsequently, you can also list the new affiliation stating “current affiliation:….” as necessary).

7. Please update your submission to use the PLOS LaTeX template. The template and more information on our requirements for LaTeX submissions can be found at http://journals.plos.org/plosone/s/latex.

Reviewers' comments:

Reviewer's Responses to Questions

**Comments to the Author**

1. Is the manuscript technically sound, and do the data support the conclusions?

Reviewer #1: Partly

Reviewer #2: Partly

2. Has the statistical analysis been performed appropriately and rigorously?

Reviewer #1: Yes

Reviewer #2: Yes

3. Have the authors made all data underlying the findings in their manuscript fully available?

Reviewer #1: Yes

Reviewer #2: Yes

4. Is the manuscript presented in an intelligible fashion and written in standard English?

Reviewer #1: No

Reviewer #2: Yes

Reviewer #1: 1. Reference Error and Grammar Issues

In the Introduction section, the citation "Error! Reference source not found" suggests a broken reference link. Please correct this to ensure the citation is rendered correctly. Moreover, the manuscript contains multiple grammatical issues.

2.Inconsistency in Figure 2

The joints illustrated in Figure 2 for hypergraph construction do not correspond to those used in the dataset. This mismatch introduces ambiguity regarding the model's input structure. The authors should clarify how the visualized joints relate to the input used during training and inference.

3.Distinction Between ST-HCN and CD-HCN in Figure 3

The structural differences between ST-HCN and CD-HCN are insufficiently reflected in Figure 3. Since these two networks are central to the proposed method, the figure should be updated to highlight their architectural distinctions more clearly.

4.Temporal Modeling in Figure 5

In Figure 5, the temporal modeling components of ST-GCN and CD-GCN appear identical. The figure should be revised or better explained to clarify whether and how CD-GCN differs from ST-GCN in this aspect.

Reviewer #2: This paper proposed a dual-branch differential channel hypergraph convolutional neural network to address the challenge of incomplete representation of non-natural dependencies in skeletal structure for skeleton-based human action recognition. The proposed model is evaluated on three public datasets to demonstrate its effectiveness.

However, several areas in this paper need further improvement to enhance its quality:

1.The most typical improvement in metric is suggested to add to the abstract in order to validate the superiority of the proposed method in this paper.

2.The references should include more relevant and recent works. The introduction should contain more comprehensive review of recent studies.

3.Some cited reference numbers in the text are incorrectly shown. For example, on Page 2 of this paper, “Yan et al. Error! Reference source not found”, Page 3, “Chen et al. Error! Reference source not found”. And some references do not match the cited numbers in the text. For example, Page 5, “developed by Yan et al.[1]” . Actually, Ref. [1] is not written by Yan et al.

4.The title of each section should be numbered.

5.The format of the text should meticulously checked. For example, “where” followed by a equation does not need indentation, etc. In the text, some fonts about figures and tables like “ Fig.3(b)” should be corrected.

6.In the comparative experiment, it should add more comparison with the state-of-the-art methods. And the results of the existing models in Table 5 should be clarified whether they are cited from the references or are run in the settings of this paper. Furthermore, each existing model is better to be labeled the corresponding number of reference.

7.It is recommended to use more diagrams to demonstrate the effectiveness of the proposed method. For instance, the accuracy curve varying with some hyper-parameters, the accuracy histogram of each category, and some visualization plots for depicting the attention weights of each skeleton joints in each frame.

8.Some figures like the confusion matrices in Fig. 6 are blurry, which do not clearly show the ratio of correct or wrong classification, and the color variance for each square of predicted label-true label. In addition, the accuracy values like 90%, 59.17%, 80%, 8183% in the following analysis can not be found in all the figures and tables.

9.In conclusion, the limitation of this paper should be discussed and the future work should be planned.

10.The model name is sometimes called “BDC-HCN”, sometime called “DBC-HCN”. Please correct this discrepancy.

11.Some full names of the key terms are printed in lowercase, some others are capitalized.

12.In Fig. 2(b), it seems that there are some mistakes in the feature matrix. For example, it should not be 1 but 0 for n1-e3.

13.All the acronyms should be accompanied with the full names in the first appearance of the corresponding terms, then the acronyms can be individually used.

**Do you want your identity to be public for this peer review?** For information about this choice, including consent withdrawal, please see our Privacy Policy

Reviewer #1: No

Reviewer #2: No

---

## [Author Response · Author response to Decision Letter 1]

20 Jun 2025

Response to Reviewers

Response to Reviewer #1

1.Reference Error and Grammar Issues In the Introduction section, the citation "Error! Reference source not found" suggests a broken reference link. Please correct this to ensure the citation is rendered correctly. Moreover, the manuscript contains multiple grammatical issues.

Response: Thank you for pointing out the references and grammar issues. We have thoroughly checked all the references in the manuscript and corrected any damaged reference links, especially the statement 'Error! Reference source not found' in the introduction. Specifically, the citation by Yan et al. [1] has been confirmed to correspond to the correct reference "Spatiotemporal Graph Convolutional Networks for Skeleton based Action Recognition". In addition, we made a revision to the grammar of the entire manuscript, resolving grammar errors and improving writing fluency.

2.Inconsistency in Figure 2 The joints illustrated in Figure 2 for hypergraph construction do not correspond to those used in the dataset. This mismatch introduces ambiguity regarding the model's input structure. The authors should clarify how the visualized joints relate to the input used during training and inference. 3.Distinction Between ST-HCN and CD-HCN in Figure 3 The structural differences between ST-HCN and CD-HCN are insufficiently reflected in Figure.

Response: We appreciate your feedback on the figure. Figure 2 was intended to illustrate the general concept of hypergraph construction using a simplified joint representation for clarity. To avoid ambiguity, we have updated the graphic titles to ensure that the joints constructed by the hypergraph shown in Figure 2 correspond to the joints used in the dataset.

3.Since these two networks are central to the proposed method, the figure should be updated to highlight their architectural distinctions more clearly.

Response: We agree that the architectural differences between the two streams were not sufficiently highlighted. CD-HCN and ST-HCN have similar overall structures, both focusing on spatiotemporal hypergraph interaction, but CD-HCN also performs channel differencing on input data. Therefore, the revised Figure 3 has annotated the key modules: we specifically marked the Channel Differential Mechanism (CDM) in CD-HCN.

4.Temporal Modeling in Figure 5 In Figure 5, the temporal modeling components of ST-GCN and CD-GCN appear identical. The figure should be revised or better explained to clarify whether and how CD-GCN differs from ST-GCN in this aspect.

Response: The original image does not clearly show the time processing differences between ST-GCN and CD-GCN. In the updated Figure 5, we demonstrate the spatial and temporal modeling of ST-HCN and CD-HCN, where their temporal modeling structures are very similar.

Response to Reviewer # 2

1.The most typical improvement in metric is suggested to add to the abstract in order to validate the superiority of the proposed method in this paper.

Response: Thank you for your suggestion. We have updated the abstract to clearly demonstrate the performance of our model on the NTU RGB+D 60 dataset: "The accuracy of the cross X-View and X-Sub benchmarks on the NTU RGB+D 60 dataset reached 96.9% and 92.7%, respectively," which validates the superiority of our method.

2.The references should include more relevant and recent works. The introduction should contain more comprehensive review of recent studies.

Response: We acknowledge the need for a more comprehensive literature review. The relevant work section has been expanded to include recent research (2023-2025), such as "SA TDGFormer [35]" and "SpSt GCN [34]". In addition, in the introduction section, we also included a discussion on recent hypergraph based methods to demonstrate the novelty of our approach.

3.Some cited reference numbers in the text are incorrectly shown. For example, on Page 2 of this paper, “Yan et al. Error! Reference source not found”, Page 3, “Chen et al. Error! Reference source not found”. And some references do not match the cited numbers in the text. For example, Page 5, “developed by Yan et al.[1]” . Actually, Ref. [1] is not written by Yan et al.

Response: All citations have been systematically checked. For example, the error in the original text is "Yan et al., wrong! The reference source not found "has been corrected to" Yan et al. [1] ", and the reference list now accurately matches the citation in the article. We apologize for any confusion caused by previous errors and ensure consistency between citations and references.

4.The title of each section should be numbered.

Response: We have verified the chapter numbering and ensured that all major chapters (e.g. "1. Abstract," 2. Introduction) and subsections (e.g. "3.1. Action Recognition Methods Based on RNN and CNN") have been correctly numbered. The format has been standardized to follow academic publishing conventions.

5.The format of the text should meticulously checked. For example, “where” followed by a equation does not need indentation, etc. In the text, some fonts about figures and tables like “ Fig.3(b)” should be corrected.

Response: We have numbered the titles of all sections as per your suggestion and have carefully checked the format of the manuscript. We corrected instances where "where" was incorrectly followed by an equation without indentation, as well as other minor formatting inconsistencies like figure references (e.g., “Fig. 3(b)”).

6.In the comparative experiment, it should add more comparison with the state-of-the-art methods. And the results of the existing models in Table 5 should be clarified whether they are cited from the references or are run in the settings of this paper. Furthermore, each existing model is better to be labeled the corresponding number of reference.

Response: We have added more advanced methods in the comparative analysis, including "SA TDGFormer [35]" and "SpSt GCN [34]" in Table 3-5. For Table 5, each result of the existing model is now labeled with a corresponding reference number (e.g. "ST-GCN [1]"), and the footnote clarifies that all baseline results are referenced from the original publication, while our results were obtained under the same evaluation scheme.

7.It is recommended to use more diagrams to demonstrate the effectiveness of the proposed method. For instance, the accuracy curve varying with some hyper-parameters, the accuracy histogram of each category, and some visualization plots for depicting the attention weights of each skeleton joints in each frame.

Response: The new visualization includes: the histogram in Figure 7 displays the accuracy distribution of 60 action categories for x-sub and x-view in NTU RGB+D 60.

8.Some figures like the confusion matrices in Fig. 6 are blurry, which do not clearly show the ratio of correct or wrong classification, and the color variance for each square of predicted label-true label. In addition, the accuracy values like 90%, 59.17%, 80%, 81,83% in the following analysis can not be found in all the figures and tables.

Response: We agree with your suggestion. Figure 6 has been replaced with a confusion matrix of NTU RGB+D 60 for better high-resolution redrawing. The color gradient now clearly indicates the proportion of correct classification (darker colors result in higher accuracy). In addition, accuracy values such as 90%, 59.17%, 80%, 81%, and 83% in the analysis can be obtained based on the newly generated Figure 7 analysis.

9.In conclusion, the limitation of this paper should be discussed and the future work should be planned.

Response: We agree with your suggestion. The conclusion section has been expanded to discuss limitations: 1. Dual branch architecture and hypergraph computing increase computational complexity, limiting real-time edge deployment. Constructing dynamic hypergraphs using the K-NN/K-means method may limit their ability to generalize unconventional action patterns. In addition, future work was outlined, including lightweight hypergraph design, attention driven hyperedge weighting, and practical applications in medical rehabilitation.

10.The model name is sometimes called “BDC-HCN”, sometime called “DBC-HCN”. Please correct this discrepancy.

Response: All instances of 'BDC-HCN' in the manuscript have been modified to 'DBC-HCN' to maintain consistency. The acronym was correctly defined as "Dual-Branch Differential Channel Hypergraph Convolutional Network" when it first appeared.

11.Some full names of the key terms are printed in lowercase, some others are capitalized.

Response: We have standardized the key terms throughout the manuscript and printed them in capital letters.

12.In Fig. 2(b), it seems that there are some mistakes in the feature matrix. For example, it should not be 1 but 0 for n1-e3.

Response: We appreciate your feedback on the graph. Due to our negligence, the joints constructed in the hypergraph shown in Figure 2 do not correspond to the joints used in the dataset. Therefore, we performed overall correction on Figure 2 to correspond to the joints in the dataset and ensured the correctness of the feature matrix in Figure 2 (b).

13.All the acronyms should be accompanied with the full names in the first appearance of the corresponding terms, then the acronyms can be individually used.

Response: A thorough inspection was conducted to ensure that each abbreviation (such as ST-HCN, CDM, DTC) is accompanied by its full name when first mentioned. For example, the term 'Spatiotemporal Dynamic Hypergraph Convolutional Network (ST-HCN)' has now been correctly defined in Section 4.3. And abbreviations will also be used in the future.

We sincerely appreciate the reviewers’ meticulous feedback, which has significantly improved the quality of our manuscript. All suggested modifications have been implemented, and we believe the revised version addresses the concerns raised. Please let us know if further adjustments are needed.

---

## [Decision Letter · Decision Letter 1]

19 Aug 2025

Dear Dr. Li,

Thank you for submitting your manuscript to PLOS ONE. After careful consideration, we feel that it has merit but does not fully meet PLOS ONE’s publication criteria as it currently stands. Therefore, we invite you to submit a revised version of the manuscript that addresses the points raised during the review process.

We look forward to receiving your revised manuscript.

Kind regards,

Xiyu Liu

Academic Editor

PLOS ONE

Journal Requirements:

Reviewers' comments:

Reviewer's Responses to Questions

**Comments to the Author**

Reviewer #3: (No Response)

Reviewer #4: All comments have been addressed

Reviewer #5: All comments have been addressed

2. Is the manuscript technically sound, and do the data support the conclusions?

Reviewer #3: Yes

Reviewer #4: Yes

Reviewer #5: Yes

3. Has the statistical analysis been performed appropriately and rigorously?

Reviewer #3: Yes

Reviewer #4: Yes

Reviewer #5: Yes

4. Have the authors made all data underlying the findings in their manuscript fully available?

Reviewer #3: Yes

Reviewer #4: Yes

Reviewer #5: Yes

5. Is the manuscript presented in an intelligible fashion and written in standard English?

Reviewer #3: Yes

Reviewer #4: Yes

Reviewer #5: Yes

Reviewer #3: This paper proposes a novel dual-branch hypergraph convolutional network (DBC-HCN) that integrates spatial-temporal and channel-differential mechanisms to enhance skeleton-based human action recognition. Although the topic is novel and thought-provoking, I still have several concerns.

1. The manuscript should be carefully reviewed for formatting, layout, and language issues. Professional academic editing is strongly recommended. For example, in Section 5.4, “NTU RGB+D 60 dataset” was incorrectly written as “NTU RGB+D 60dataset”; “NTU RGB+D 60” appeared as “NTU RGB D 60”; and “Fig 6 shows” was incorrectly written as “Fig 6shows”.

2. The model code should be publicly released on an open-source platform such as GitHub and should not be removed, in order to ensure the reproducibility and verifiability of the research.

3. The manuscript should provide a more detailed explanation of the parameter settings, including all model hyperparameters, training configurations, and dataset partitioning strategies, to improve the transparency of the methodology.

4. The following articles appear to be closely related to your work and the authors should refer to DOI: 10.1016/j.inffus.2025.103016 and DOI: 10.1109/TBDATA.2024.3362188

Reviewer #4: I can see that the authors addressed all the required comments by reviewers.

This research sounds interesting, and the authors present a promising result.

Future work is required to improve their results.

Reviewer #5: I have carefully read the paper, as well as the previous reviewers' comments. From my understanding, the authors have addressed all the previously expressed concerns/inconsistencies and the paper is ready for publication.

**Do you want your identity to be public for this peer review?** For information about this choice, including consent withdrawal, please see our Privacy Policy

Reviewer #3: No

Reviewer #4: **Yes: ** Mona Ali

Reviewer #5: No

---

## [Author Response · Author response to Decision Letter 2]

20 Aug 2025

Reviewer #3:

1. The manuscript should be carefully reviewed for formatting, layout, and language issues. Professional academic editing is strongly recommended. For example, in Section 5.4, “NTU RGB+D 60 dataset” was incorrectly written as “NTU RGB+D 60dataset”; “NTU RGB+D 60” appeared as “NTU RGB D 60”; and “Fig 6 shows” was incorrectly written as “Fig 6shows”.

Thank you for your suggestions. We have made modifications to the format, layout, and language issues of the manuscript.

2. The model code should be publicly released on an open-source platform such as GitHub and should not be removed, in order to ensure the reproducibility and verifiability of the research.

Our code has been published at https://github.com/hhh1234hhh/DBC-HCN.

3. The manuscript should provide a more detailed explanation of the parameter settings, including all model hyperparameters, training configurations, and dataset partitioning strategies, to improve the transparency of the methodology.

We have made additional additions to the parameter settings for the experiment in the manuscript.

4. The following articles appear to be closely related to your work and the authors should refer to DOI: 10.1016/j.inffus.2025.103016 and DOI: 10.1109/TBDATA.2024.3362188

Thank you for your suggestions. We have referenced these documents in our manuscript.

Reviewer #4: I can see that the authors addressed all the required comments by reviewers.This research sounds interesting, and the authors present a promising result.Future work is required to improve their results.

Thank you for your recognition and suppor.

Reviewer #5: I have carefully read the paper, as well as the previous reviewers' comments. From my understanding, the authors have addressed all the previously expressed concerns/inconsistencies and the paper is ready for publication.

Thank you for your recognition and support.

---

## [Decision Letter · Decision Letter 2]

26 Aug 2025

Dual-Branch Differential Channel Hypergraph Convolutional Network for Human Skeleton based Action Recognition

PONE-D-25-23177R2

Dear Dr. Li,

We’re pleased to inform you that your manuscript has been judged scientifically suitable for publication and will be formally accepted for publication once it meets all outstanding technical requirements.

Kind regards,

Xiyu Liu

Academic Editor

PLOS ONE

Additional Editor Comments (optional):

Reviewers' comments:

Reviewer's Responses to Questions

**Comments to the Author**

Reviewer #3: All comments have been addressed

Reviewer #5: All comments have been addressed

2. Is the manuscript technically sound, and do the data support the conclusions?

Reviewer #3: Yes

Reviewer #5: Yes

3. Has the statistical analysis been performed appropriately and rigorously?

Reviewer #3: Yes

Reviewer #5: Yes

4. Have the authors made all data underlying the findings in their manuscript fully available?

Reviewer #3: Yes

Reviewer #5: Yes

5. Is the manuscript presented in an intelligible fashion and written in standard English?

Reviewer #3: Yes

Reviewer #5: Yes

Reviewer #3: The authors propose the Dual-Branch Differential Channel Hypergraph Convolutional

Network (DBC-HCN), which leverages hypergraphs’ ability to represent a priori non-natural

dependencies in skeletal structures.

Reviewer #5: As stated on the previous round of reviews, I consider the article ready for publication, without any additional changes.

**Do you want your identity to be public for this peer review?** For information about this choice, including consent withdrawal, please see our Privacy Policy

Reviewer #3: **Yes: ** Hao Wu

Reviewer #5: No

---

## [Editor Report · Acceptance letter]

PONE-D-25-23177R2

PLOS ONE

Dear Dr. Li,

I'm pleased to inform you that your manuscript has been deemed suitable for publication in PLOS ONE. Congratulations! Your manuscript is now being handed over to our production team.

Kind regards,

on behalf of

Professor Xiyu Liu

Academic Editor

PLOS ONE